# Global Health Resource Utilization and Cost-Effectiveness of Therapeutics and Diagnostics in Immune Thrombotic Thrombocytopenic Purpura (TTP)

**DOI:** 10.3390/jcm12154887

**Published:** 2023-07-25

**Authors:** Ayesha Butt, Cecily Allen, Adriana Purcell, Satoko Ito, George Goshua

**Affiliations:** 1Department of Internal Medicine, Yale School of Medicine, New Haven, CT 06510, USA; 2Division of Hematology, Department of Medicine, Johns Hopkins University, Baltimore, MD 21218, USA; 3Yale School of Medicine, New Haven, CT 06510, USA; 4Section of Hematology, Department of Internal Medicine, Yale School of Medicine, New Haven, CT 06510, USA

**Keywords:** TTP, thrombotic thrombocytopenic purpura, ADAMTS13, treatment, diagnosis, review, cost, cost-effectiveness, health resource utilization, decision science

## Abstract

In this review, we examine the current landscape of health resource utilization and cost-effectiveness data in the care of patient populations with immune thrombotic thrombocytopenic purpura. We focus on the therapeutic (therapeutic plasma exchange, glucocorticoids, rituximab, caplacizumab) and diagnostic (ADAMTS13 assay) health technologies employed in the care of patients with this rare disease. Health resource utilization and cost-effectiveness data are limited to the high-income country context. Measurement of TTP-specific utility weights in the high-income country context and collection of health resource utilization data in the low- and middle-income country settings would enable an evaluation of country-specific quality-adjusted life expectancy and cost-effectiveness of these therapeutic and diagnostic health technologies. This quantification of value is one way to mitigate cost concerns where they exist.

## 1. Introduction

Thrombotic thrombocytopenic purpura (TTP) is a rare and life-threatening thrombotic microangiopathy characterized by a triad of microangiopathic hemolytic anemia, severe thrombocytopenia, and organ ischemia linked to disseminated, microvascular platelet rich-thrombi [1]. Acute episodes of TTP are marked by a severe deficiency of a disintegrin and metalloproteinase with a thrombospondin type 1 motif, member 13 (ADAMTS13), with the standard of care initial diagnosis being defined by enzymatic activity of less than 10% of normal ADAMTS13 in countries where this assay is available [2,3].

Decreased ADAMTS13 activity leads to an accumulation of ultra-large von Willebrand factor multimers, which bind platelets and induce aggregation [1]. Most commonly, a severe deficiency of ADAMTS13 is mediated by the immune-mediated production of autoantibodies to ADAMTS13. In one cross-sectional study of 772 patients with a first episode of thrombotic microangiopathy and ADAMTS13-confirmed diagnosis of TTP, 73% of patients were noted to have positive anti-ADAMTS13 IgG, with an additional 3% being identified beyond the first episode [4]. In this same study, 21% of patients were always anti-ADAMTS13 IgG-negative and 3% had mutation-confirmed (homozygous or compound heterozygous) congenital TTP [1]. The epidemiology of TTP, whether immune-mediated or hereditary, classifies it as a rare disease in every country in the world, being defined, at its most strict, as <650 patients per one million people in Brazil [5]. The estimated annual incidence in different studies is geographic-location-dependent and ranges between 1 and 13 cases per million people, while the prevalence of TTP is 10 per million people [4,6,7].

The first case of what would later be termed “TTP” was reported by Eli Moschcowitz in 1924 [8]. In 1959, the first known documented utilization of whole blood exchange transformed an 11-year old girl’s “deep and persistent” coma to the regaining of consciousness and the disappearance of abnormal neurologic signs [9]. In 1977, two patients were reported to recover completely after “intensive plasmapheresis” [10], paving the way for treatment transformation in 1991 through a clinical trial of therapeutic plasma exchange versus plasma infusion that spanned the 1980s [11]. In 1982, the presence of ultra-large von Willebrand Factor (VWF) multimers in four patients with chronic, relapsing TTP suggested a defect in the processing of these multimers after secretion [12]. In 1985, large amounts of VWF (when compared with fibrinogen) were found within the visceral platelet microthrombi via histopathology in a deceased TTP patient [1]. In 1991, the aforementioned clinical trial of therapeutic plasma exchange versus plasma infusion reported an acute survival probability of 96% versus 84%, with 6-month survival holding at 78% and 63%, respectively. These results were a significant increase from a mortality of *up to* 90% prior to plasma infusion or exchange [1,11]. A novel metalloprotease (VWF cleaving-protease) that specifically cleaved VWF was purified from human plasma in 1996, and two years later, a severe functional deficiency of the VWF cleaving-protease was shown to cause TTP [13,14]. Finally, this cleaving-protease was identified as ADAMTS13 in 2001, setting the stage for a biomarker-based diagnostic assay [15].

The treatment of immune TTP is anchored around therapeutic plasma exchange, which serves to remove autoantibodies and ultra-large von Willebrand factor multimers, and to replenish ADAMTS13 [11]. Autoantibody formation is targeted by using immunosuppression with a foundation of glucocorticoids and rituximab [16]. In 2019, caplacizumab, an anti–von Willebrand factor humanized single-variable-domain immunoglobulin (Nanobody, Ablynx) that targets the A1 domain of von Willebrand factor, obtained Food and Drug Administration approval for the management of TTP based on a primary endpoint of time to platelet normalization [17]. Standard-of-care treatment in high-income countries revolves around the aforementioned therapeutics, noting that the addition of glucocorticoids to therapeutic plasma exchange is a strong recommendation and that the addition of rituximab and caplacizumab are conditional recommendations in the 2020 International Society on Thrombosis and Haemostasis Treatment Guidelines [18], with a grade IB recommendation against using caplacizumab in unselected patients in the 2022 American Society of Hematology Education Program [19]. Treatment is initiated based on clinical suspicion in patients for evidence of thrombotic microangiopathy on smear and without a more likely etiology, employing either the PLASMIC and/or French scores [20]. In this primer, we examine the current landscape of health resource utilization and cost-effectiveness studies of the therapeutic armamentarium and diagnostic assay for patients with this rare disease. Specifically, we focus on TPE, glucocorticoids, rituximab, caplacizumab, and the ADAMTS13 assay, with a primary focus on health resource utilization and a secondary one on cost-effectiveness.

## 2. Therapeutic #1: Therapeutic Plasma Exchange (TPE)

TPE is an established anchor of frontline treatment for immune TTP, improving acute and 6-month mortality even when compared to plasma infusion [1,11,21]. TPE should be started as soon as the diagnosis of TTP is suspected [1,3]. Daily TPE removes autoantibodies against ADAMTS13 while replenishing the missing or inhibited ADAMTS13 enzyme [11]. Treatment with TPE is continued daily until the platelet count is greater than 150 × 10^9^/L on two consecutive days, and TPE may be performed more than once daily in severely ill patients [22,23,24].

***Health resource utilization:*** A 2008 retrospective analysis that used the South East England TTP Registry, spanning a time period from April 2002 through to December 2006 across 176 patients with 236 acute episodes, reported a median of 15.5 plasma exchanges (range 3–93), with a significant difference between the number of exchanges required in the first acute episode of TTP (median 15 (4–40)) compared to that in relapsed acute TTP episodes (median 9.5 (5–57)) [25]. The authors noted a statistically significant secular trend—in this case, a systematic decrease over time, presumably due to increased clinician experience—within their 56-month study period, with median exchange sessions decreasing from 19 to 12 between the first 20 and last 36 months. A 2016 analysis of 2009–2014 data from the Australian TTP/TMA registry reported similar findings, with TPE used in 67/72 (93%) episodes of 57 confirmed TTP patients with a median of 12 (range 1–8) TPE sessions per acute episode [26].

A 2012 study using data from the HealthCore Integrated Research Database (HIRD), explored healthcare utilization of immune TTP patients in a commercially insured population from many different regions of the United States [27]. The study included 151 patients with the mean total healthcare payments for the TTP hospitalization being USD 56,347 (standard deviation [SD] USD 80,230) [27]. During an acute episode of TTP, the mean number of PE was found to be 8.5 (SD 12.9, range 1–116). The mean duration of the PE treatment was 28.2 days (SD 54.0) [27]. Mean payments for TPE services in the month following discharge were USD 9127 (SD USD 20,840) [27].

A 2014 study including results from a survey of physicians from 32 centers in the US sought to delineate current clinical practices of TTP treatment in the US [28]. At all centers, TPE was started as soon as TTP was the most likely diagnosis [28]. TPE was initiated with plasma as replacement fluid (91%) at 1.0 plasma volume (72%) and stopped with a platelet count of 150 × 10⁹/L (66%), and TPE was then tapered off (69%) [28].

In 2021, an American retrospective study utilized administrative claims data between 2010 and 2018 for US Medicare and non-Medicare populations following immune TTP episodes to describe immune TTP-related hospital resource utilization, cost, complications, and overall survival [29]. The study included 2279 patients at a weighted mean age of 58 across four payer types: Medicare Fee-for-Service (FFS; n = 1486), Medicare Advantage (MA; n = 123), commercial (n = 312), and Managed Medicaid (MM; n = 358). The study reported a mean length of hospitalization across payer types ranging between 12 and 16 days and 61% of Medicare FFS patients received ICU-level care. Notably, Medicare FFS patients experienced a mean of 3 days from admission to TPE initiation, as compared to all other payers for whom TPE initiation occurred less than 1 day from admission. [29] Mean total direct healthcare expenditures for index hospitalization varied by payer (Medicare FFS: USD 29,024; MA: USD 12,860; commercial: USD 9996; MM: USD 10,470) [29]. The FFS cohort had mortality rates of 15.7% and 21% during initial hospitalization and within the first 30 days, respectively [29].

***Cost-effectiveness:*** At this time, there are no cost-effectiveness studies regarding TPE in immune TTP.

## 3. Therapeutic #2: Steroids

Steroids have long been considered an important therapeutic option for immune TTP, owing to the autoimmune nature of the disease. Although no longer treated with corticosteroids alone, at one point in the pre-ADAMST13 time period, corticosteroids alone were reported to lead to remission in patients with milder cases [30,31]. While there is a paucity of randomized controlled trials exploring the use of steroids in immune TTP, several studies have reported favorable outcomes leading to its use as part of standard of care for TTP alongside TPE [23].

***Health resource utilization:*** In a 2006 Italian randomized open-label trial of 60 TTP patients running from 2000 through 2006, high-dose methylprednisolone (10 mg/kg/day for 3 days followed by 2.5 mg/kg/day) induced a higher complete remission rate by a pre-specified day 23 compared to standard-dose methylprednisolone (1 mg/kg/day) (76.6% vs. 46.6%, respectively) [32]. Costs of care between these two arms were not reported. Another randomized clinical trial examining the adjunct use of prednisone versus cyclosporine to TPE was halted after interim analysis (22 total evaluable patients) showed an increase in exacerbations with the latter strategy during acute immune TTP treatment, in addition to prednisone significantly improving ADAMTS13 activity at weeks 3 and 4 and lowering anti-ADAMTS13 IgG at weeks 2–4 compared to cyclosporine [33]. The 2008 retrospective analysis from the South East England TTP Registry showed that in 80% of all acute episodes, TTP patients received steroids. Most patients received three days of pulsed therapy between 500 mg and 1 g per day of methylprednisolone for up to 3 days [25]. Data from the Australian TTP/TMA registry similarly showed that steroids were used in 71% (51/72) acute TTP episodes with a median steroid exposure being 19.5 days (range 2–108 days) [26]. A 2014 study including results from a survey of physicians from 32 centers in the US reported the standard use of prednisone for TTP treatment in 26/32 centers (81%) [28].

***Cost-effectiveness:*** At this time, there are no cost-effectiveness studies on corticosteroid use in immune TTP.

## 4. Therapeutic #3: Rituximab

Rituximab is a chimeric monoclonal antibody directed against the CD20 antigen present on B lymphocytes. It is used in the management of diseases like lymphoma and rheumatoid arthritis, where it clears B lymphocytes responsible for antibody production via complement-dependent cytotoxicity, antibody-dependent cellular cytotoxicity, or directly by inducing apoptosis [34]. The role of rituximab in treating patients with TTP emerged first in the refractory or relapsed setting in the 2000s [35,36,37,38].

***Health resource utilization:*** A 2009 randomized clinical trial to evaluate the addition of upfront rituximab to TPE and steroids was stopped due to low enrolment [39]. A prospective, non-randomized, single-arm, phase 2 trial reported in 2011 by the South East England TTP study group examined the benefit of upfront rituximab with steroids and TPE versus historical control patients treated without rituximab and with steroids and TPE. The investigators reported a reduction in relapse rates from 57% at a median 18 months in the historical controls to 10% at a median 27 months in 40 trial patients with the use of weekly rituximab at lymphoma-based dosing (375 mg/m^2^ weekly for 4 weeks, up to 8 infusions allowed), administered within three days of admission in the acute TTP setting [40]. Across 34 total patients classified as de novo TTP, the authors noted the following three ethnicity categories and their respective, descriptive median number of TPE treatments and range: Afro-Caribbean (24, range: 6–34), White 11.5 (range 4–30), and Indian/Asian 23 (range, 6–34). They then collapsed ethnicity into a binary variable of white versus nonwhite, reporting the number of TPE sessions to be significantly decreased at 14 versus 21, respectively [40]. A study using the United States Thrombotic Microangiopathies Consortium iTTP Registry explored the correlation of race with mortality and relapse-free survival (RFS) in immune TTP in the United States from 1995 to 2020. Black race was associated with shorter RFS (hazard ratio [HR], 1.60; 95% CI, 1.16–2.21), while the addition of rituximab to corticosteroids improved RFS in White (HR, 0.37; 95% CI, 0.18–0.73) but not Black patients (HR, 0.96; 95% CI, 0.71–1.31) [41].

Subsequent to the 2009 study, a 2012 retrospective study from the United Kingdom TTP Study Registry reported the administration of rituximab within 3 days from admission in 54 patients with acute de novo TTP versus rituximab administration more than 3 days from admission in 32 patients with acute de novo TTP over a 2004–2011 time period [42]. In this context, the former strategy yielded earlier remission (12 versus 20 days), fewer TPE sessions (16 versus 24) and a shorter length of hospital admission (16 versus 23 days) [42]. In 2012, the expert-based guidelines from the United Kingdom recommended the utilization of upfront rituximab alongside TPE and steroids for patients with relapsed TTP and suggested consideration for upfront rituximab in the de novo context [43]. Data from the first 5 years of the Australian TTP/TMA microangiopathy registry reported rituximab use in 28/72 (39%) acute TTP episodes spanning the 2009–2014 time period. The authors noted the first registry-documented use of rituximab in Australia in 2010, with its use occurring in 21/52 (40%) and 7/20 (35%) of de novo and relapsed TTP episodes in this time period [26]. A 2019 systematic review and meta-analysis reported on a total of 570 patients across nine included studies dating from 2011 through to 2018 [44]. The odds ratio for the pooled relapse rates as an effect of rituximab administration in the acute phase was 0.40 (95% confidence interval (CI) 0.19–0.85, I^2^ = 43%) with an OR of 0.41 ([95% CI 0.18–0.91], I^2^ = 0%) from the six and eight studies, respectively, where these data were reported [44]. A 2014 study including results from a survey of physicians from 32 centers in the US reported heterogeneity in the use of rituximab: 4 centers (13%) did not use rituximab, while 28 centers (88%) used it routinely [28]. Of the 28 centers that used rituximab, 5 (18%) routinely used the drug during the first presentation of TTP, while the remaining 23 centers (82%) use it in relapsed/refractory cases [28]. Rituximab was used concurrently with TPE in 27/28 centers (96%) [28].

In the United States, one major hurdle to adding rituximab into inpatient treatment paradigms for immune TTP is the expense of the medication [45], particularly in the context of a payer system that delineates fixed payments per pre-specified diagnosis-related group and thus incentivizes cost-containment in the inpatient setting. To examine this question, a retrospective single-center study of 27 patients with de novo immune TTP across all relapses over a median follow-up time of 56 months reported on preventable TPE sessions, inpatient hospital days, and cost savings that would have accrued for patients who responded to the addition of incident rituximab therapy to TPE and steroids after subsequent relapses [46]. First in the initial cohort, rituximab use during initial hospitalization was projected to avert 185 inpatient admission days and 137 TPE sessions at cost savings to hospitals of USD 900,000 [46]. Second in the relapse cohort, rituximab use during any earlier relapse was projected to avert 86 inpatient admission days and 64 TPE sessions at cost savings to hospitals of USD 420,000 [46]. This study delineated that, despite financial inpatient formulary concerns, the inpatient use of rituximab in the management of TTP can lead to cost savings (~USD 30,000 per patient with TTP initiated on rituximab inpatient). This supported the practice change to an automatic formulary approval for inpatient rituximab use for all patients with immune TTP in the same hospital health system [46].

***Cost-effectiveness:*** At this time, there are no cost-effectiveness studies on rituximab use in immune TTP.

## 5. Therapeutic #4: Caplacizumab

Caplacizumab is a humanized monoclonal antibody that binds to vWF, inhibits its interaction with platelet glycoprotein Ib-IX-V, yields rapid suppression of vWF-ristocetin cofactor activity, and prevents platelet adhesion [47]. In 2018 and 2019, caplacizumab was approved by the European Medicines Agency and the Food and Drug Administration based on the time to platelet normalization for the treatment of immune TTP in conjunction with TPE and immunosuppression [47]. In 2020, the International Society on Thrombosis and Haemostasis (ISTH) guidelines for the management of immune TTP noted a conditional recommendation for the use of caplacizumab in the frontline treatment of immune TTP [18].

***Health resource utilization:*** Healthcare utilization with the addition of caplacizumab use to TPE and immunosuppression versus without caplacizumab was descriptively reported in a phase 3, randomized, two-arm clinical trial of patients. Here, with caplacizumab use, the patients (1) received a median of 5 days (range 1–35) of plasma exchange (without: 7.0; range 3–46), (2) underwent a median of 18 L (range 5–102) plasma exchanged (without: 27; range 4–254), (3) had a median 9-day (range 2–37) length hospital of stay (without: 12; range 4–53), with (4) a median 3-day (range 1–10) stay in the intensive care unit (without: 5; range 1–47). The median number of TPE sessions and hospital length of stay observed with caplacizumab use in real-world data from Germany, France, England, and Spain are 9 (range 2–41) and 18 (range 5–79), 5 (IQR 4–7) and 13 (IQR 9–19), 7 (IQR 5–14) and 12 (IQR 8–24), and 8.5 (IQR 6–12.5) and 12 (IQR 9–15), respectively [16,19,48,49,50]. Of note, healthcare utilization data from trials and real-world data of caplacizumab do not apply to immune TTP patients deemed to be high bleeding risk patients, as these individuals are noted to be excluded in all studies to date. A 2022 systematic review and meta-analysis of caplacizumab use examined data from 632 patients across phase 2 and phase 3 clinical trials, two observational studies with historical controls (total n = 175 treated with caplacizumab with n = 219 historical controls), and one observational study with a concurrent control (total n = 18), with all three observational studies reported as having a high risk of bias [51]. The work had a literature search censor date through July 2021. Noting caveats regarding an inability to adjust for secular trends and confounding as a limitation of historical control arms and the exclusion of high-risk bleeding patients, the authors noted that caplacizumab added to the standard of care versus the following standard of care alone yielded relative risks: (1) RR 1.37 ([CI 1.06, 1.77]) and (RR 7.10 [CI 0.90, 56.14]) for all bleed risk in clinical trial and observational data, respectively; and (2) RR 0.21 [CI 0.05, 1.74]) and 0.62 [CI 0.07, 4.41] for death in clinical trial and observational data, respectively. Focusing on the International Society on Thrombosis and Haemostasis major bleed rate, while again noting the exclusion of patients deemed at high risk of bleeding from receiving caplacizumab, the lowest cumulative per-patient-treated real-world data estimate for ISTH major bleeding with caplacizumab use is 2.4% (Table 1). Included within this, with caplacizumab use through to December 2022, there were 4 intracranial hemorrhages that occurred in 272 immune TTP episodes over a median follow-up of 80 days across English, Spanish, and Austrian experiences [49,50,52]. In contrast, there were 0 intracranial hemorrhages across 219 patients over a median follow-up of 618 days and 0 intracranial hemorrhages over a median follow-up of 2336 days in consecutively treated patients with immune TTP in the Harvard TTP Collaborative and Oklahoma TTP Registry, respectively [53,54].

A Sanofi-funded cost analysis (note: not cost-effectiveness) projected healthcare utilization expected with caplacizumab use in US hospital setting based on the phase 3 trial and peer-reviewed literature, with a payer mix of 20% Medicare Fee-for-Service (FFS), 8% Medicare Advantage, 10% Medicaid, and 62% commercial [56]. Specifically, the authors reported a per-patient cost of the standard of care as USD 23,000 versus USD 70,000 for caplacizumab added to the standard of care, with these values being inclusive of projected savings of USD 8000 and USD 15,000 for length of stay and TPE cost-savings, respectively, and yielding an incremental per-patient cost of USD 47,000 with caplacizumab use. The authors note that when considering patients cared for under the Medicare Fee-for-Service model specifically, the incremental per-patient cost with caplacizumab would decrease to USD 5000 when the 2020 Medicare new technology add-on payment (NTAP) for caplacizumab is applied. This is an important consideration in the short run that also does not address the long-term perspective given the 3-year expiration of indication-specific NTAPs in the Medicare Fee-for-Service context.

Cost concerns are also noted in France, where caplacizumab is approved for use in immune TTP [57]. The authors note that the addition caplacizumab per immune TTP patient, at an assumed per-11mg caplacizumab dose cost of EUR 4400, would increase costs three-fold per immune TTP episode treated from a historic baseline of EUR 75,000. With a per-11 mg caplacizumab dose cost of USD 8800, the authors also postulate a 4-fold increase in the United States for the same [57].

***Cost-effectiveness:*** Cost is one concern with the use of caplacizumab, which has been examined in several cost-effectiveness analyses [58,59,60]. In December 2020, after initial rejection, a series of meetings and re-appraisal of Sanofi’s modeling at their fourth meeting, the National Institute for Health and Care Excellence (NICE) officially recommended caplacizumab use with TPE and immunosuppression for patients with immune TTP [58]. Specifically, NICE noted that Sanofi’s new modeling reported an ICER of just under GBP 30,000 per quality-adjusted life year (QALY) for caplacizumab use, an upper threshold (the typical is GBP 20,000 per QALY) NICE can employ when considering contextual factors such as innovation. This led to their recommendation of caplacizumab as a cost-effective option for treating immune TTP, paving the way for its use in the National Health Service [58]. Concerns about input parameters and assumptions employed by Sanofi in their final model have now been documented [61]. In no specific order, these include the following: (1) the use of an acute mortality relative risk of 0.5; (2) unsupported utility values—values which reflect quality of life—such as a lifelong 34% quality of life reduction in living in remission if a patient achieved remission without caplacizumab use as compared to the same remission with caplacizumab use; (3) an underestimated annual relapse rate that over a lifetime time-horizon would propagate a significant artificial decrease in the ICER for their drug [61].

A 2020 American cost-effectiveness analysis presented the results of two decision tree models, one for each clinical trial, and a third Markov model with a 5-year time-horizon conducted from the US health system perspective and at a willingness-to-pay threshold of 195,000 USD/QALY [60]. The analysis eschewed the consideration of bleeding—increased with caplacizumab use—and employed utilities preset to minimize the incremental cost-effectiveness ratio for caplacizumab added to the standard of care versus standard of care alone. At a 5-year time-horizon the incremental cost for caplacizumab was USD 400,000 with an incremental QALY of 0.27, resulting in an ICER of USD 1.5 million, far exceeding the willingness-to-pay threshold. In the American context, caplacizumab was reported to be cost-ineffective in 100% of 10,000 iterations in a probabilistic sensitivity analysis with deterministic sensitivity analysis identifying drug price as the one parameter that could significantly decrease the ICER, beginning at a price reduction of at least 80% [60,61].

A 2021 Italian health technology assessment conducted a cost-effectiveness analysis of capalcizumab added to the standard of care versus the standard of care at a lifetime time-horizon, an Italian hospital perspective and at a willingness-to-pay threshold of 60,000 EUR/QALY previously suggested in Italy [59]. The authors reported an expected increase in 3.27 life years and 3.06 quality-adjusted life years with caplacizumab use at an incremental cost-effectiveness ratio of 44,600 EUR/QALY. These results are accrued in the context of acute mortality probabilities of 0.0% with caplacizumab use and 13.2% without caplacizumab use per each immune TTP across a lifetime time-horizon. Deterministic sensitivity analysis did not evaluate a non-0% probability of acute mortality with caplacizumab use and the probabilistic sensitivity analysis reported the probability of caplacizumab being cost-effective at 82.4% without discounts for 10-mg caplacizumab per-vial pricing at EUR 3900 in the Italian setting.

In the context of a 2022 open-label single-arm phase 3 trial (MAYARI, NCT 05468320) to examine the use of caplacizumab without frontline TPE in immune TTP with a planned 1-year follow-up, a cost-effectiveness analysis examined the efficacy threshold of a 1-year relapse-free survival this single-arm trial would need to meet for caplacizumab to be a cost-effective replacement of the standard of care [62]. The analysis employed clinical trial data, assuming caplacizumab without TPE is exactly as effective as caplacizumab with TPE and with all inpatient costs eliminated. A relapse-free survival of 100% at 1-year would not meet any accepted US willingness-to-pay threshold. This analysis also entirely ignored bleeding complications, assumed no drop-off in efficacy from opting for no TPE therapy and reported that a minimum 78% price decrease in caplacizumab would be needed. TPE-based care was the cost-effective strategy in 100% of 10,000 iterations in a probabilistic sensitivity analysis [62].

A sample state transition diagram for model-based cost-effectiveness analysis is shown in Figure 1.

A comparative summary of the costs, advantages, disadvantages, and side effects for the four therapeutics can be found in Table 2. Of note, costs vary greatly across and within countries; thus, we largely focus on the largest payer in the United States, the Centers for Medicare and Medicaid Services (Table 2).

## 6. Diagnostics

Testing for ADAMTS13 activity requires laboratory skill and time, making in-house and same-day assay output challenging. Clinical scoring systems are free of charge and include the French and PLASMIC scores. Neither score has perfect test characteristics and this worsens with older age [70]. Due to these limitations, there is a demand for rapid diagnostic testing and, given the associated costs, a need for cost-effectiveness analyses to convince hospital stakeholders that investment is worthwhile.

***Health resource utilization:*** ADAMTS13 activity assay availability was limited in the early 2000s and the options included immunoradiometric, collagen-binding, and ristocetin-cofactor assays [71,72,73,74]. The availability of testing has since expanded, with some simplification achieved in testing complexity. Most current assays for ADAMTS13 utilize fluorescence resonance energy transfer (FRET) to detect proteolysis of vWF [75], improving the turnaround time of results in the context of reduced testing complexity observed with other assays [76]. Calls for further testing simplification led to the development of fully automated assays. The HemosIL AcuStar ADAMTS13 activity assay is a fully automated, chemiluminescent assay that takes 33 min for analysis output [77]. Studies comparing the HemosIL chemiluminescent immunoassay to chromogenic ELISA and FRET-based assays revealed high concordance in results [77,78,79,80,81]. The HemosIL Acustar ADAMTS13 activity assay may then decrease the burden placed on laboratories to run skill- and time-intensive assays and increase the turnaround time of ADAMTS13 testing. The latter would help decrease unnecessary risk exposure of patients suspected to have immune TTP to TTP-specific therapeutic modalities that each have associated risks. A cost analysis (note: not cost-effectiveness) compared what the theoretical costs in 2019 would have been at a single center if an AcuStar was used instead of the ELISA-based Technozym ADAMTS13 [82]. From ELISA-based testing of 165 patient plasma samples across 93 patients received in that year for ADAMTS13 activity testing, 9 patients underwent unnecessary TPEs. The authors reported this corresponded to 18 TPE sessions and 227 units of fresh frozen plasma with a total cost of EUR 41,700. While the counterfactual clinical outcome cannot be known and the authors did not employ a measure of effectiveness, there is an implied clinical benefit of avoiding unnecessary patient risk exposure to TPE. Compared to the 2019 analysis year, in this 2022 study, the availability of caplacizumab and its attendant bleeding risk puts patients at unnecessary increased risk exposure for bleeding when diagnostic assay turnaround times are prolonged.

***Cost-effectiveness:*** Two cost-effectiveness analyses conducted by the same group have examined the comparative value of in-house versus send-out FRET-based ADAMTS13 activity (Table 3) [64,83]. The first in 2017 reported on four competing testing strategies for a patient suspected of having immune TTP: (1) send-out ADAMTS13 activity versus (2) in-house ADAMTS13 activity versus (3) PLASMIC clinical score application followed by send-out ADAMTS13 testing for intermediate-to-high risk PLASMIC-scoring patients versus (4) PLASMIC clinical score application followed by in-house ADAMTS13 testing for intermediate-to-high risk PLASMIC-scoring patients [83]. The respective 3-day costs and deaths averted (the effectiveness outcome the authors employed) were USD 15,600 and 0.98, USD 4900 and 0.91, USD 9400 and 0.95, and USD 4700 and 0.91, respectively. Judged at a threshold of <USD 50,000 per death averted, the authors noted strategy #2 was the cost-effective strategy at an ICER of 49,600 per death averted. In a 2020 cost-effectiveness analysis, the authors examined the impact of delayed FRET-based ADAMTS13 activity testing over a 6-day time-horizon with a focus on the number of days delayed [64]. They reported an incremental cost to the health system ranging from USD 4200 to USD 5100 for every 1-day delay in obtaining ADAMTS13 activity assay results, with life days, their metric of effectiveness, decreasing from 5.89 with a 1-day delay to 5.85 with a 5-day delay.

## 7. Conclusions

In 2023, health resource utilization data and cost-effectiveness evaluation of therapeutics and diagnostics in the care of patients with immune TTP are preferentially available in high-income country settings. Despite this, in high-income countries, the measurement of quality-adjusted life expectancy and cost-effectiveness evaluation is limited by the lack of TTP-specific utility weights. In low- and middle-income countries, these limitations are compounded by the lack of health resource utilization data, impaired access to the standard of care therapeutics, and the lack of ADAMTS13 assay availability due to cost concerns [84,85]. Although health technology assessments in country-specific contexts would quantify the value of each therapeutic and diagnostic in the care of patients with immune TTP, the measurement of input parameters and their uncertainty is needed for such quantification. For this reason, we believe the pressing research needs include the direct measurement of TTP-specific utility weights for high-income country settings and the collection of health resource utilization data in low- and middle-income country settings. 

## Figures and Tables

**Figure 1 jcm-12-04887-f001:**
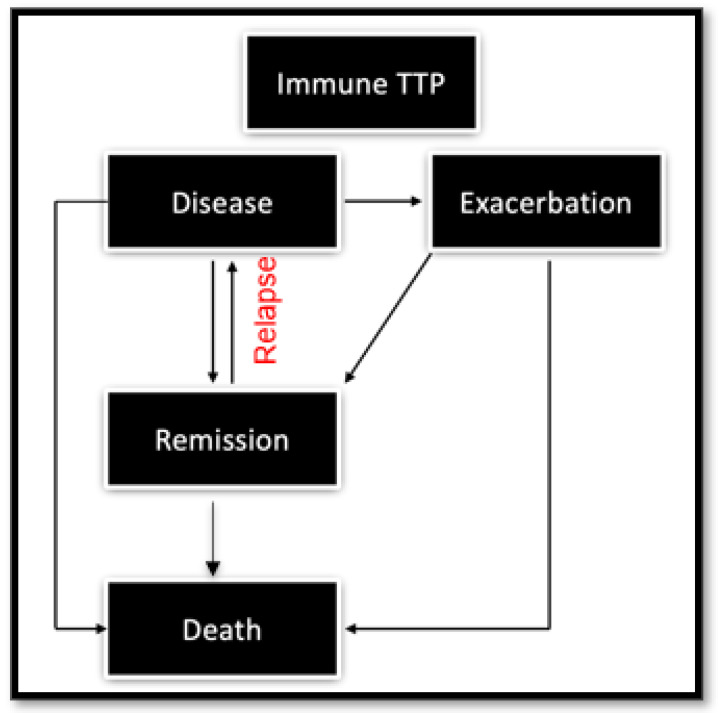
Sample state transition diagram for modeling treatment decisions in immune TTP [62]. Shown are the various health states possible for patients with TTP, which include the disease state (i.e., active TTP necessitating treatment), the remission state (no active TTP), exacerbation state and death. In this sample, relapse is a transitory event. All patients start in the disease state, with possibility of exacerbation, remission, and relapse back to disease. At all disease states, patients accrue background mortality risk, in addition to mortality specific to a given disease state.

**Table 1 jcm-12-04887-t001:** ISTH major bleeding reported in real-world data (RWD), with and without caplacizumab, for patients with immune TTP [55].

Parameter	Caplacizumab RWD (Knöbl et al. [52])	Caplacizumab RWD (Pascual Izquierdo et al. [50])		Caplacizumab RWD (Coppo et al. [16])	Caplacizumab RWD (Dutt et al. [49])		Caplacizumab RWD (Volker et al. [48])
**Sample size**	20	77		90	85		60
**ISTH Major** **Bleed # (%)**	1 (5%)	Not reported		2 (2.2%)	5 (5.9%)		Not reported
**Parameter**	***Without* Caplacizumab** **RWD (Harvard Collaborative [53])**		***Without* Caplacizumab** **RWD (Dutt et al. [49])**			***Without *Caplacizumab** **RWD (Page et al. [54])**	
**Sample size**	219		39			68	
**ISTH Major** **Bleed # (%)**	0		0			1 (1.5%) (CVC line placement)	

**Table 2 jcm-12-04887-t002:** Comparative summary of selected costs, advantages, disadvantages, and side effects of the therapeutic plasma exchange, steroids, rituximab, and caplacizumab. Legend: CMS = Centers for Medicare and Medicaid Services; mg = milligram.

Therapies	Therapeutic Plasma Exchange (TPE)	Steroids	Rituximab	Caplacizumab
Cost per CMS unit	Average cost per 1 TPE session: USD 4900 Cost range: USD 3000–USD 6600 [56,60,63,64]	USD 0.02/mg of prednisone [65] USD 5.95/125 mg of methylprednisone [65]	USD 81.75 per 10 mg [65]	USD 718.64 per mg [65]
Advantages	Total of 30 years of data, Lifesaving [11]	Globally available, low cost	Total of 20 years of data; associated with increased time to relapse [66] in the short-run; can be cost-saving to hospitals in the long-run [46]	Easy administration, associated with decreased exacerbations [3]
Disadvantages	Requires logistical coordination; not available at all hospitals	Need to minimize unnecessary patient steroid exposure (i.e., overly prolonged tapers)	Recent observational study reporting differential effect of the social construct of race on relapse-free survival [41]	Associated with longer time to ADAMTS13 recovery [67], increased relapses [3], increased major bleeding [19], high cost [19]
Side Effects	Fever, urticaria, and hypocalcemic symptoms [68]	Immunosuppression leading to increased risk of infections; osteoporosis, myopathy, increase in blood glucose levels, weight gain, hypertension, arrhythmias, skin atrophy, acne, mild hirsutism, stria, impaired wound healing [69]	Immunosuppression leading to increased risk of infections; rituximab hypersensitivity; serum sickness and exacerbation of cardiac symptoms [42]	Bleeding (Table 1) [19]

**Table 3 jcm-12-04887-t003:** ADAMTS13 cost-effectiveness analyses with noted effectiveness metrics, time-horizon, and thresholds [64,83].

Effectiveness Metric and Time-Horizon	Strategies and/or Comparators	Effectiveness|Cost	Cost-Effective Strategy| at Threshold
Deaths averted and 3-day time-horizon [64]	Send-out vs. In-house vs. PLASMIC → send-out vs. PLAMSIC → in-house	0.98|USD 15,600 0.91|USD 4900 0.95|USD 9400 0.91|USD 4700	In-house (USD 49,600 per death averted), at threshold: <USD 50,000 per death averted
Life days and 6-day time-horizon [83]	Delay to ADAMTS13 results: day x vs. day x + 1, where x = 1 − 4	1 Day Delay: 5.89|USD 27,500 5 Day Delay: 5.85|USD 46,500	Not applicable

## Data Availability

Not applicable.

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
