# Peer review of "Global Health Resource Utilization and Cost-Effectiveness of Therapeutics and Diagnostics in Immune Thrombotic Thrombocytopenic Purpura (TTP)"

_jcm, 2023, doi:10.3390/jcm12154887_

Round 1

Reviewer 1 Report

The manuscript by Butt and colleagues reports data on cost-effectiveness of diagnostic and therapeutic tools routinely used in the management of patients affected by thrombotic thrombocytopenic purpura.

Overall, the topic is of interest for clinicians treating this kind of patients, considering the elevated costs of recently approved drugs such as caplacizumab and upcoming drugs such as recombinant ADAMTS-13. However, I believe such a topic should be reported as a Letter to the Editor or as a brief Report to a clinicians’ audience, instead of as an original article.

I think the interest of this manuscript can be restricted to the two paragraphs on the cost-effectiveness analysis of caplacizumab use and ADAMTS13 activity test.

Reviewer 2 Report

Interesting article on a rare topic

Quite informative article style

A Markov model that can be used / included in HTAs for treatment evaluation is missing

At least a comparative summary table of the various treatments is needed (assumed cost in dollars, advantages, disadvantages, side effects)

Reviewer 3 Report

Introduction gives definition and good review of history of TTP.  Would include the current standard of diagnosis and standard of care of treatment (could specify in the U.S. since this is likely different in other parts of the world where there is limited access to newer treatments and/or even treatments such as rituxan).  Since this is a cost effectiveness discussion, I think it would be helpful to state the general costs of standard treatments (i.e. HD catheter placement to initiate TPE, cost per unit of FFP, cost of steroids, cost of testing, cost of course of treatment with rituxan etc).  I would include in the introduction that treatment is started based on clinical suspicion in patients with TMA on smear, and unlikey to have another more likely cause (DIC, medications etc).  

In line 38 "whether immune mediated or not...."  Agree that acquired and hereditary TTP are both rare diseases but would reword this as treatment is different.  Would also specify acquired TTP in title as it seems that is where this paper is focused.  

Would add a table including cost effectiveness of ADAMTS13 testing (line 373) to better visualize.  A table comparing the cost and cost effectiveness (where available) of current standard treatment for aTTP would also be helpful.  

Overall, I think this is a great discussion and topic and enjoyed reading it and and important consideration in newer treatments.   

Reviewer 4 Report

Excellent review which includes some cost data from high resource and low resource settings.  i have no comments/suggestions about the content presented.  Very well referenced.

A few minor semantic points:

"Primer" usually refers to an introduction of a topic in order to educate- this is a higher-level analysis.  Can title be changed?

I had to look up "secular" as I thought it applied to non-spiritual things, but I now see it has an economic meaning as well (not changing over time).  Maybe defining it once on first use would be helpful for the readers.

As above

Round 2

Reviewer 1 Report

No other comments.

Reviewer 2 Report

The authors enriched the paper according to my suggestions. In any case, the paper has its own clinical value even if the topic is health economics

Reviewer 4 Report

I appreciate the authors response to review and modifications made to protocol.